# Preliminary Investigation of Different Types of Inoculums and Substrate Preparation for Biohydrogen Production

**Bidattul Syirat Zainal** [1,*], **Sabrina Zaini** [2], **Ali Akbar Zinatizadeh** [3], **Nuruol Syuhadaa Mohd** [2], **Shaliza Ibrahim** [4], **Pin Jern Ker** [1,5] and **Hassan Mohamed** [1,6]

1 Institute of Sustainable Energy, Universiti Tenaga Nasional, Jalan IKRAM-UNITEN, Kajang 43000, Selangor, Malaysia
2 Department of Civil Engineering, Faculty of Engineering, University of Malaya, Jln Profesor Diraja Ungku Aziz, Seksyen 13, Kuala Lumpur 50603, Selangor, Malaysia
3 Department of Applied Chemistry, Faculty of Chemistry, Razi University, Kermanshah 6714414971, Iran
4 Institute of Ocean & Earth Sciences, University of Malaya, Jln Profesor Diraja Ungku Aziz, Seksyen 13, Kuala Lumpur 50603, Selangor, Malaysia
5 Electrical and Electronics Engineering Department, College of Engineering, Universiti Tenaga Nasional, Jalan IKRAM-UNITEN, Kajang 43000, Selangor, Malaysia
6 Mechanical Engineering Department, Universiti Tenaga Nasional, Jalan IKRAM-UNITEN, Kajang 43000, Selangor, Malaysia
* Correspondence: syirat88@gmail.com

**Abstract:** A pre-culture stage is required to obtain robustly-dividing cells with high hydrogen ($H_2$) production capabilities. However, a step-by-step process for biohydrogen production is scarcely reported, mainly from palm oil wastewater. Therefore, this study developed a guideline to find the best inoculum heat treatment conditions and implement the selected conditions for biohydrogen production using palm oil wastewater. This study used raw palm oil mill effluent (POME) and POME sludge as substrate and inoculum, respectively. Our findings reveal that 80 °C and 30 min were the best conditions for inoculum heat treatment. When testing the conditions on POME sludge and inoculating with raw POME (28 g COD/L) at 37 °C (reaction temperature), 24 h (reaction time), and pH 5.5, 34 mL $H_2$/d was recorded. A slight increase (1.1-fold) was observed compared to 5 g COD/L POME co-digested with 5 g/L glucose (31 mL $H_2$/d). This discovery indicates that raw POME is a potential source for biohydrogen production under anaerobic fermentation and can be directly used as substrate up to 30 g COD/L. The proposed guideline could also be implemented for different organic wastes for biohydrogen production study.

**Keywords:** anaerobic digestion; biohydrogen; heat treatment; palm oil; renewable sources; energy

## 1. Introduction

Hydrogen ($H_2$) is considered a clean and long-lasting energy carrier. For the past 15 years, the focus on hydrogen production from organic waste has been increasing, fuelling research in the field. Organic wastes, such as agricultural waste, contain specific microbes that may contribute to hydrogen production in the future and prove to be a viable renewable energy source. Increasing the performance of biohydrogen production from lignocellulosic biomass is thus an important research direction since biohydrogen is a green and environmentally friendly energy carrier that has the potential to reduce our reliance on fossil fuels.

Palm oil mill effluent (POME) is one such source of agricultural waste, and palm oil milling is one of Malaysia's most important industries. POME is produced by milling either dry or wet palm oil fruit bunches. Several studies reported that 5 to 7.5 tonnes of water are used for every metric tonne of crude palm oil (CPO) produced, with more than half of this typically ending up as POME [1,2]. Generally, POME contains a high amount

of organic nutrients and different suspended materials. Hence, if it is not treated appropriately, POME will have a crucial environmental impact [3].

Biological treatment/processes such as dark fermentation and photo-fermentation are other methods to produce biohydrogen. The former uses anaerobic bacteria, while the latter requires phototropic bacteria. The use of mixed cultures as inoculum for biohydrogen production via dark fermentation has recently gained much attention [4–6]. Several biohydrogen studies utilizing POME under the dark fermentation process have been reported in a batch [1] and continuous mode [7].

The differences between dark and photo-fermentation are that the former does not require light, wastewater could be used as a substrate, carbon dioxide is released along with hydrogen generation, and acidic pH favours hydrogen production. In contrast, photo-fermentation does not require light, toxic and high-strength wastewater cannot be utilized as substrate, carbon dioxide will be fixed to a carbon source along with hydrogen generation, and a near-neutral pH is preferred during the process [8]. In the dark fermentation process, several mesophilic and thermophilic microbial species, such as *Enterobacter cloacae*, *Clostridium butyricum*, *Bacillus amyloliquefaciens*, and *Thermoanaerobacterium*, break down waste material into hydrogen [8].

Other important factors, such as the pH in the bioreactor, temperature, substrate source, and the pre-treatment of a substrate or/and inoculums, must be considered to achieve successful biological treatment for biohydrogen production using organic wastes [9]. Inoculum pre-treatment involves heat shock, acid/base, chemical, and physical pre-treatment [10]. Introducing an optimization strategy would significantly affect biohydrogen production regarding the required factors and conditions.

Magrini et al. studied the effect of different inoculum heat treatments on biohydrogen production and volatile fatty acids from sugarcane vinasse as substrate at pH 6 [10]. They reported that the highest hydrogen yield and production of 4.75 mmol $H_2$ $g^{-1}$ COD and 821.34 mL, respectively, were achieved when the inoculum was heat treated at 90 °C for 10 min. They also confirmed and proved that no supplements are needed during the process, and this process could be driven only by pH selection and inoculum pre-treatment.

Meanwhile, Mohan et al. investigated a rapid and straightforward strategy for assessing biohydrogen production potential (BHP) using mixed cultures from composite wastewater as a biocatalyst and different wastewaters as a substrate [11]. Under critical operational factors, the designed strategy demonstrated the feasibility of using selected wastewater for biohydrogen production. They also found that pre-treatment of anaerobic inoculum is associated with acidic feeding conditions and significantly affects overall substrate degradation and hydrogen production, regardless of the types of wastewater employed. The nature/composition of the wastewater and the applied organic load also significantly impacted process efficiency.

Therefore, based on the factors described above, optimization strategies for different inoculum heat treatments are one of the main parameters that must be investigated. In addition, a specific environment must be formed to grow hydrogen-producing bacteria (HPB). In this study, heat-shock pre-treatment was selected, as it can suppress homoacetogens in the sludge and allows the growth of HPB [12]. Furthermore, an acidic condition is favoured, as it is the best condition for effective biohydrogen production [13,14]. Thus, a preliminary investigation of different types of inoculums and substrate preparation was conducted to achieve the highest biohydrogen production.

## 2. Materials and Methods

### 2.1. Inoculum

In this study, only two different sludges were used as inoculums: (1) Carlsberg sludge (collected from an anaerobic fermentation process treating brewery wastewater in Shah Alam, Selangor, Malaysia) and (2) POME sludge (from the anaerobic pond at Jugra Palm Oil Mill, Banting, Selangor, Malaysia) (Figure 1). Food and beverage (F&B) sludge was not used, as it could not produce biohydrogen in our preliminary study. Instead, Carlsberg and POME sludge were used in this study, as they could enhance biohydrogen production by treating POME [1,12].

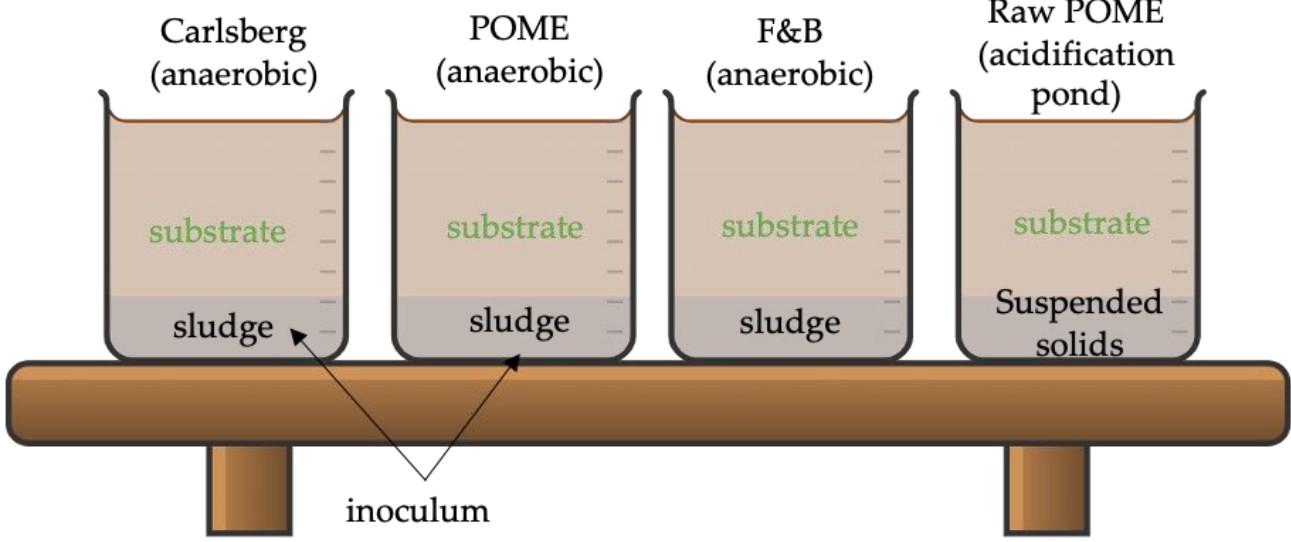

**Figure 1.** Inoculum and substrate preparation.

Samples collected were immediately stored in a cold room (4 °C). Before the experiments, they were left untreated to equilibrate at room temperature (37 °C) and allowed to settle before both sludges were heat-treated and used as inoculum. The characteristics of the sludges were measured before the experiment and are presented in Table 1.

**Table 1.** Characteristics of the inoculums and substrates used in this study.

| Parameters | Inoculum | | Substrates | | | |
|---|---|---|---|---|---|---|
| | Carlsberg Sludge | POME Sludge | Carlsberg | POME | F&B | Raw POME |
| Initial Chemical Oxygen Demand (COD) (g/L) | - | - | 12.5 | 13 | 2.5 | 28 |
| Initial pH | 7.2 | 7.3 | 7.2 | 6.9 | 7.0 | 4.9 |
| Total Suspended Solid (TSS) (g TSS/L) | 35 | 50 | - | - | - | - |
| Volatile Suspended Solid (VSS) (g VSS/L) | 24 | 35 | - | - | - | - |
| Moisture Content (%) | n. d | n. d | n. d | 80 [1] | n. d | 95–96 [1] |
| Bacterial Identification [2] | n. d | *Lactobacillus acidophilic* (Gram-positive facultative anaerobe) | - | - | - | - |
| Plate Count (CFU/mL) [2] | n. d | $2.4 \times 10^7$ | - | - | - | - |

[1] [15]; [2] [16] (n.d = not determined).

## 2.2. Substrate

Dark fermentation/anaerobic digestion requires organic materials to be converted to simple monomers such as fatty and amino acids and monosaccharides. The substrates are (1) Carlsberg, (2) POME, (3) F&B, and (4) raw POME. Before the experiment, all the wastewater was left untreated and allowed to settle before the liquid part was used to identify its feasibility as a substrate for biohydrogen production (Figure 1). The characteristics of the substrates used in this study were also summarised in Table 1.

## 2.3. Preliminarily Investigation Experiment

As shown in Figure 2, the best conditions were optimized based on the one-factor-at-a-time (OFAT) method. During the First Phase, Carlsberg sludge was heat-treated at different times and temperatures. Next, the best conditions from the First Phase (time, temperature) were selected for the Second Phase. In this phase, different inoculums and substrates were evaluated for biohydrogen production. In the Third Phase, different substrate concentrations were studied using the best heat-treated inoculum (from the Second Phase). Finally, the optimum conditions were based on the best substrate concentration and heat-treated inoculum. The details of each phase are described below.

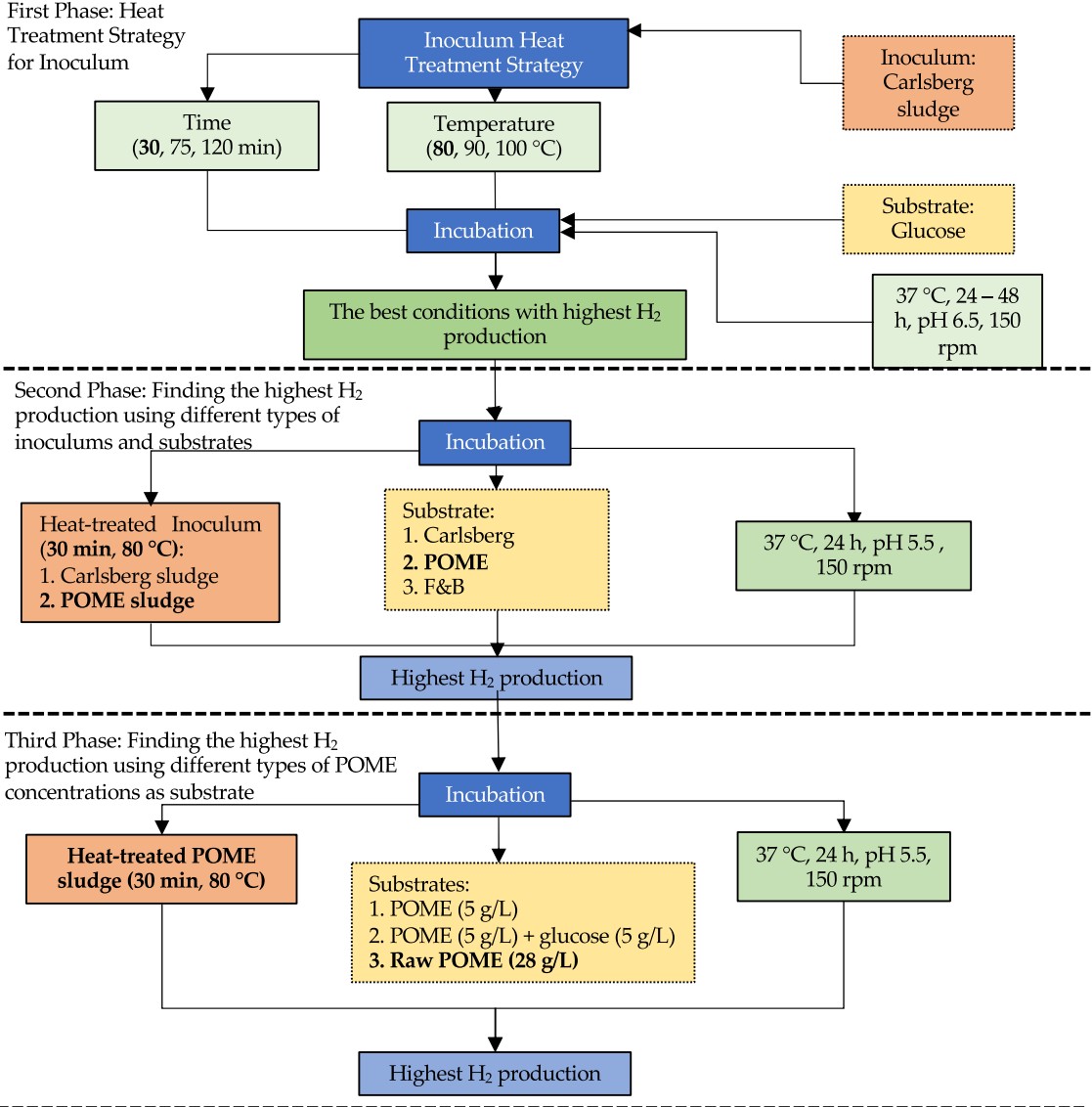

**Figure 2.** A summary of a step-by-step optimization strategy used in this study for biohydrogen production potential. ($H_2$ = biohydrogen).

### 2.3.1. First Phase: Inoculum Heat Treatments

Carlsberg sludge was heated at 80, 90, and 100 °C for 30, 75, and 120 min using a water bath. Temperatures between 80 and 100 °C between 30 and 120 min were selected based on studies done by Lin et al. (2011) for 80 °C, 30 min [17], Uyub et al. (2017) for 80 °C, 120 min and 100 °C, 30 min [18], and Woo and Song (2010) for 100 °C, 120 min [19]. Meanwhile, a heat treatment at 90 °C for 75 min was selected as the median between 80 and 100 °C and 30 and 120 min. The inoculum was then left to cool to room temperature (27 °C). Carlsberg sludge was measured for total suspended solids (TSS) according to the APHA Standard Methods 2540 G [20], then diluted to 10 g/L TSS. Glucose as substrate was added at 6 g/L, and the pH was adjusted to pH 6.5. A 156 mL serum bottle was used as a batch reactor, with a working volume of 100 mL and a headspace volume of 18 mL at 1 atm. The serum bottles were sealed using an aluminium cap before sparging with nitrogen gas at 10 mL/min for 15 min to achieve anaerobic conditions. Samples were then incubated at 37 °C at 150 rpm for 48 h (reaction time) in a shaking incubator to act as a control (Figure 3). Finally, hydrogen was collected and measured every 24 h.

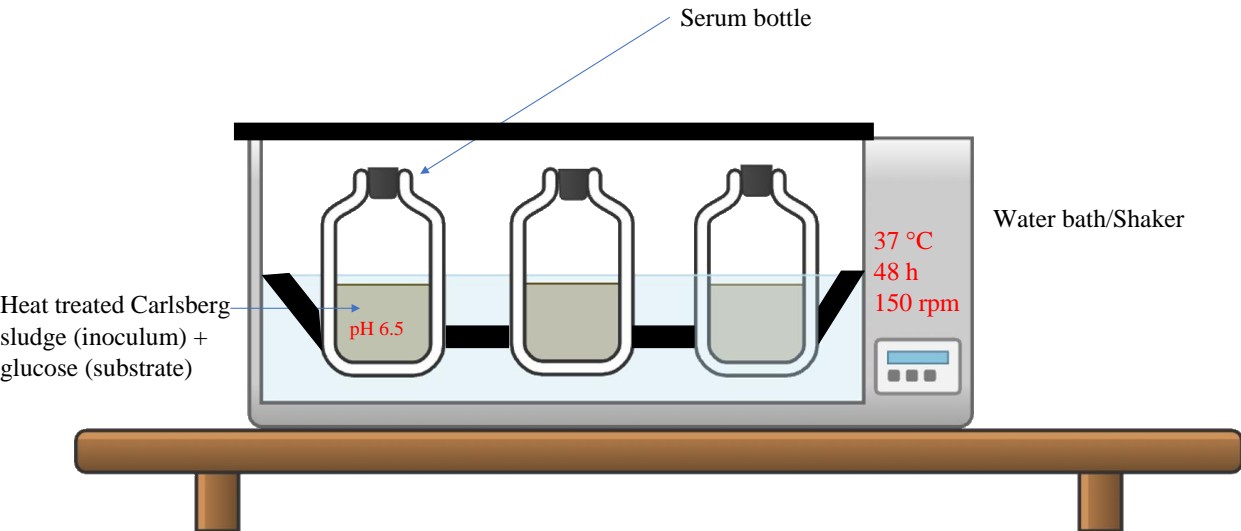

**Figure 3.** A schematic diagram depicting the incubation of cultures for biohydrogen production during the First Phase.

### 2.3.2. Second Phase: Feasibility Study Using Different Inoculums and Substrates

Next, Carlsberg and POME sludges were heat-treated based on the selected heat treatment strategy (80 °C, 30 min) to study the best inoculum for biohydrogen production). However, since the final pH of all samples in the First Phase was reduced to pH 5.5 ± 0.1, a buffer solution was added to maintain the pH of 5.5. Carlsberg and POME sludges were then inoculated (37 °C, 24 h, 150 rpm) into three substrates, as described in Figure 2, with an inoculum concentration of 10 g/L and a total working volume of 100 mL. The substrates' initial COD of Carlsberg, POME, and F&B were 12.5 g/L, 13 g/L, and 2.5 g/L, respectively. Finally, the total biohydrogen volume was measured to find the best inoculum and substrate for biohydrogen production.

### 2.3.3. Third Phase: Feasibility Study Using Different Substrate Concentrations

Subsequently, to study the effect of substrate types and concentrations on biohydrogen production, 5 g COD/L of POME, 5 g COD/L POME co-digested with 5 g/L glucose, and 28 g COD/L of raw POME were incubated with the best heat-treated inoculum found in the Second Phase. This phase aimed to determine the most suitable substrate for producing the highest amount of biohydrogen using heat-treated POME sludge as inoculum. Raw POME was chosen at high concentration because the finding could be used to justify

whether a pre-treatment is needed in the future. After incubating the cultures for 24 h at 37 °C, pH 5.5, and 150 rpm, the total biohydrogen volume was measured.

### 2.4. Analytical Method

The COD, VSS, and TSS were analyzed at the start of the experiment. All the tests were conducted according to APHA Standard Methods 5220 D, 2540 G, and 2540 D, respectively [20]. The volume of biogas produced was measured using the water displacement method [21], while volatile fatty acid (VFA) analysis was performed according to our previous study [16]. The composition of biogas was analyzed using gas chromatography (GC) (Perkin Elmer, AutoSystem Gas Chromatograph, 600 Series LINK), a pack GC column Supelco, 40/80 carboxen 1000, MR2924D, 10′ × 1/8′, and a thermal conductivity detector (TCD). At a flow rate of 30 mL min$^{-1}$, the carrier gas used was argon of high purity. The oven, injector, and detector temperatures were set to 100 °C, 150 °C, and 200 °C, respectively. For injection-related gas sampling, a 0.5 mL, 2500 μL, gas-tight syringe from Hamilton, United States, was used [1]. In addition, 1 N of hydrochloric acid (HCl) and 1 N of sodium hydroxide (NaOH) were used for pH adjustment.

## 3. Results

### 3.1. First Phase: Inoculum Heat Treatment

Figure 4A,B show that a heat treatment strategy of 80 °C for 30 min produces the highest total average biohydrogen production rate and VFA concentration over 24 (36.32 mL $H_2$/d and 64.82 mg acetic acid/L) and 48 (18.70 mL $H_2$/d and 64.54 mg acetic acid/L) hours, with initial pH of 6.5. The VFA concentration increases at 48 h for a heat treatment strategy of 80 °C for 30 min. This change demonstrates the production of VFA during hydrolysis [22], supported by the reduction in final pH of 5.5 ± 0.1. The same trend was observed for all studied conditions except for 100 °C for 120 min. This inoculum heat treatment condition shows the presence of hydrogen gas only after 48 h of reaction time; hence, no biohydrogen production rate was recorded in the first 24 h. Meanwhile, based on Figure 4C, the highest average COD removal efficiency was 52.55% and 21.94% (100 °C, 30 min), and 25.09% and 12.76% (100 °C, 120 min) after 24 and 48 h with low VFA concentration detected (<100 mg acetic acid/L). For 80 °C for 30 min, the COD removal efficiency was 38.75% (24 h) and 36.89% (48 h). The lowest COD removal efficiency was observed at inoculum heat treatment of 100 °C, 120 min (25.09% for 24 h, 12.76% for 48 h).

It is vital to clarify that no consistent patterns correlate temperature and duration of heat treatment to biohydrogen production. It is clear, however, that biohydrogen production occurs better with an initial pH of 6.5. Nonetheless, because the cultures in these experiments were not buffered, the final pH was lower due to the presence of VFAs. Therefore, further repetitions were done to confirm our findings, where the final pH was recorded and found to be pH 5.5. The result of this heat treatment strategy was then used for further experiments using the best heat treatment conditions, i.e., 80 °C for 30 min, with initial pH of 5.5, which was supported by other findings on optimal pH [12]. Phosphate buffers were then added to stabilize the pH to 5.5 to study its effect on biohydrogen production.

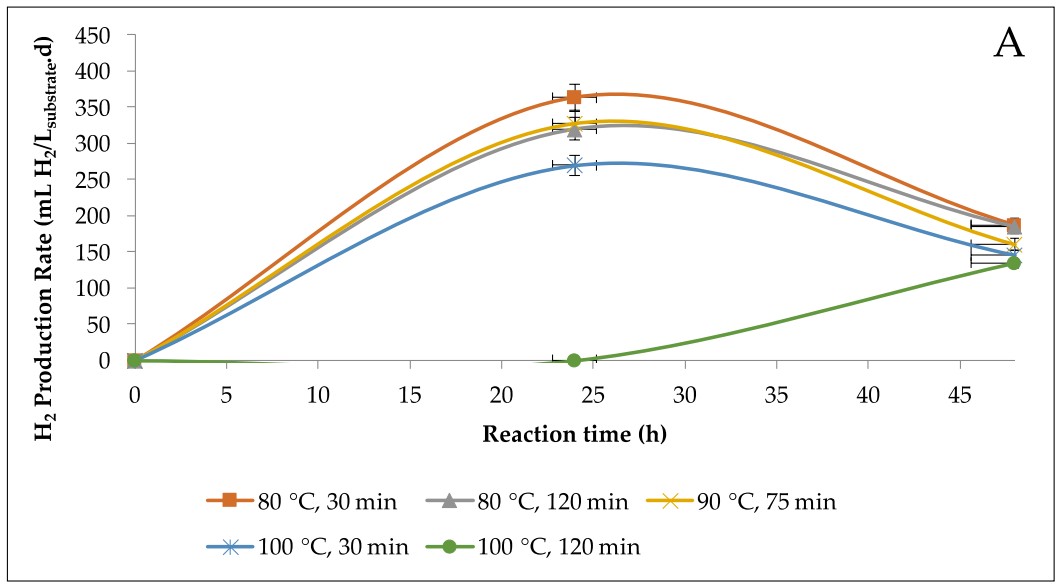

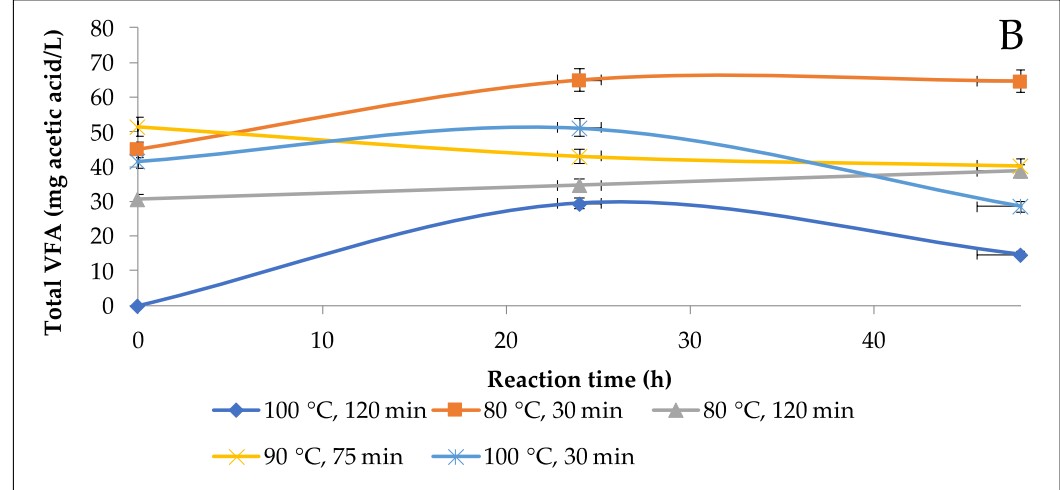

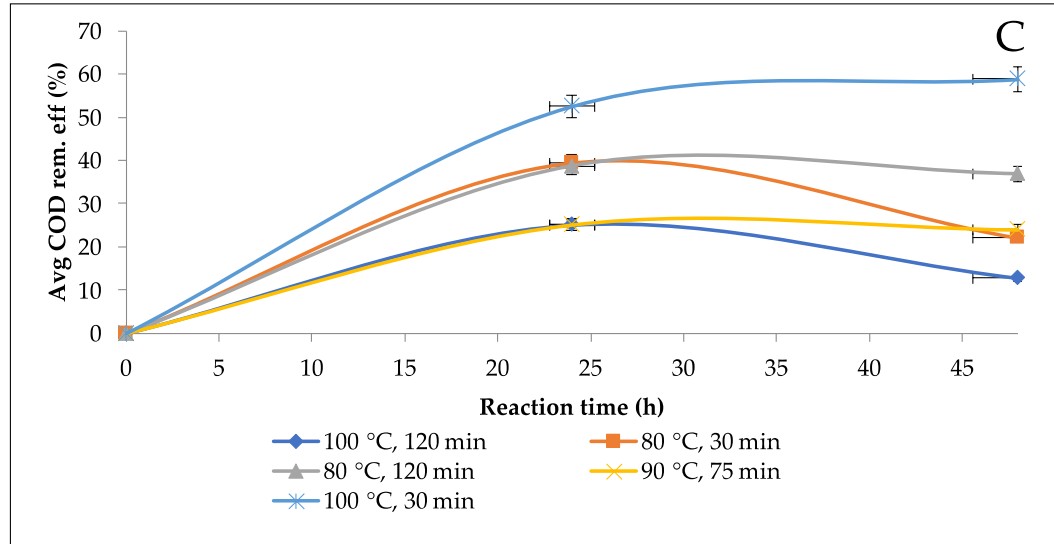

**Figure 4.** Results from heat treatment strategy (First Phase) using Carlsberg anaerobic granulated sludge as inoculum fed with 6 g COD/L glucose. (**A**) Average hydrogen production rate versus time. (**B**) Average total VFA versus time. (**C**) Average COD removal efficiency versus time. (Average results were calculated based on sample size, *n* = 3).

### 3.2. Second Phase: Effects of Different Inoculums and Substrates on Biohydrogen Production Rate

After selecting the best heat treatment conditions with the highest biohydrogen production rate, the most potential inoculum and substrate concentrations for optimal biohydrogen production were determined. POME and Carlsberg sludge were heat treated at 80 °C for 30 min and incubated at pH 5.5, 37 °C, and 150 rpm for 24 h. The substrates used are shown in Figure 1. Meanwhile, based on Figure 5, Carlsberg and POME sludges show a significant increase when grown in POME compared to Carlsberg and F&B. POME sludge shows the highest average biohydrogen production rate of 46.03 mL $H_2$/d compared to the Carlsberg sludge (34.07 mL $H_2$/d) when using POME as a substrate.

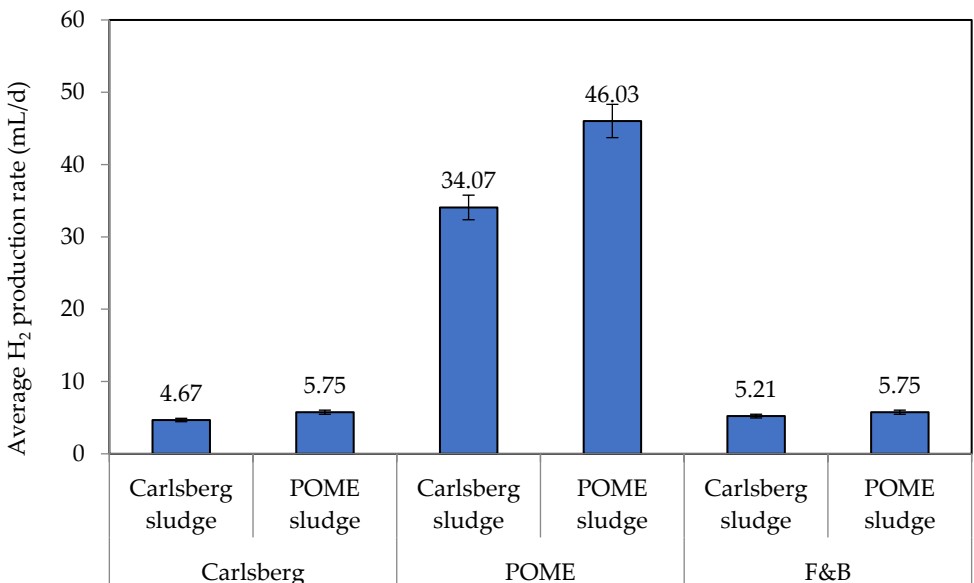

**Figure 5.** Biohydrogen produced by inoculum is derived from different sludges grown in various wastewater media (substrate) with Carlsberg = 12.5 g COD/L, POME = 13 g COD/L, and F&B = 2.5 g COD/L. The media were adjusted to pH 5.5. Experiments were done in triplicates (*n* = 3), and samples were taken at 24 h.

Meanwhile, using Carlsberg and POME sludge to produce hydrogen from Carlsberg and F&B as substrates did not generate significant amounts of biohydrogen, although they had different COD concentrations. Carlsberg and POME sludge yielded 4.67 and 5.75 mL $H_2$/d, respectively when fed with Carlsberg (substrate, 12.5 g COD/L), while 5.21 and 5.75 mL $H_2$/d were achieved when fed with F&B (substrate, 2.5 g COD/L).

### 3.3. Third Phase: Effects of Different Substrate Concentrations on Biohydrogen Production Rate

Subsequently, since POME sludge produced the highest hydrogen in the Second Phase, it was selected and grown for 24 h in various concentrations and co-digested with glucose to investigate the effects of different concentrations on the biohydrogen production rate. The substrates were 5 g/L POME, 5 g/L POME co-digested with 5 g/L glucose (POME + Glu), and 28 g/L raw POME. Results of additional experiments done in POME to study the effect of organic carbon content on hydrogen production are shown in Figure 6. The figure shows that at 5 g/L COD of POME, 10 g/L POME (including 5 g/L added glucose), and 28 g/L raw POME, POME sludge yields 9, 31, and 34 mL $H_2$/d, respectively.

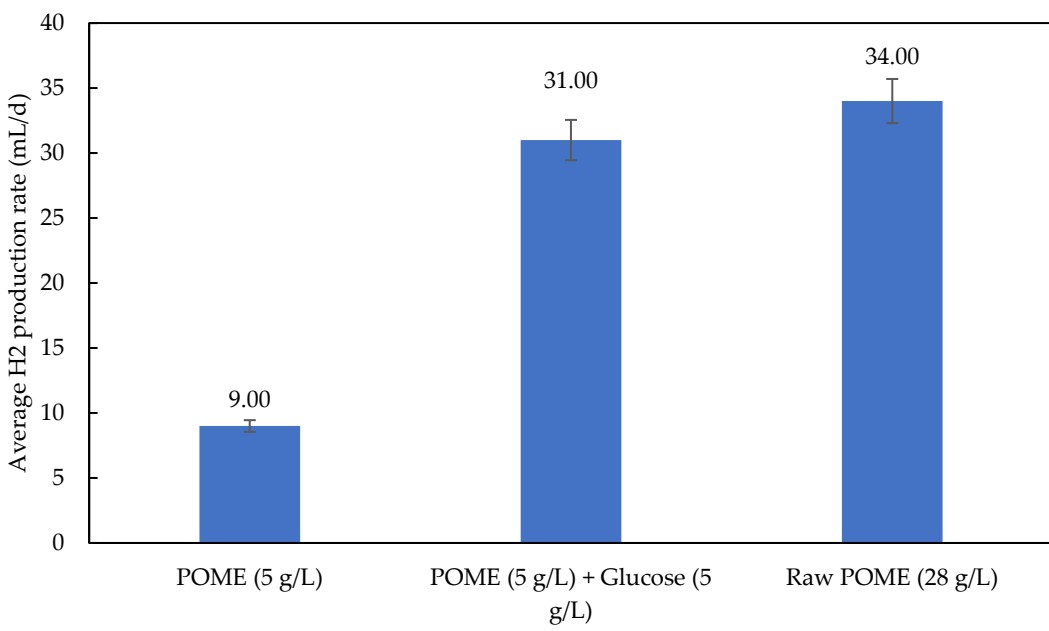

**Figure 6.** Effect of various POME concentrations on biohydrogen production rate. The cultures were grown at pH 5.5, 37 °C, 150 rpm, and 24 h reaction time. Results display error bars with a 5% value (95% confidence level).

## 4. Discussion

Inoculum heat treatment is the most used method for biohydrogen production; hence, this technique can enrich spores that form HPB [23]. A similar finding was reported by Noike et al., where suppression of lactic acid bacteria (LAB), *Lactobacillus* sp., following heat treatment for 30 min at a temperature between 50 and 90 °C resulted in higher biohydrogen production [24]. The concept of heat treatment is that certain bacteria can detect changes in their surroundings. This condition enables them to sporulate, especially at high temperatures. It is also believed that the HPB can be enriched and grown under suitable conditions, increasing biohydrogen production [25].

Moreover, the COD removal efficiency for all heat-treated sludge was reported to be between 10 and 50% after 48 h of reaction time. A similar finding was observed using heat-treated Carlsberg sludge for biohydrogen production [12]. The authors reported that the low efficiency is because the carbonaceous matter is oxidized to organic acids rather than entirely mineralized for methane and carbon dioxide, as occurs when anaerobic digestion is complete.

Since the maximum biohydrogen production was reported at pH 5.5, this study is consistent with the previous finding [26]. pH value is vital in VFA bio-production because it regulates the activities of various microbes involved in anaerobic digestion [27]. This condition helps specific microbes to have enough energy to generate biohydrogen. pH range between 5 and 5.5 may also reduce methanogenic activity while increasing the activity of HPB [26]. HPB is also responsible for VFA production as a by-product of biohydrogen production [28]. Gerardi (1979) reported that acetic, butyric, and propionic acids (short-chain fatty acids) indicate the presence of saprophytic bacteria that break down the organic matter and convert them into simpler compounds [29]. According to Pachapur et al. (2019), inoculum heat treatment would generate only hydrogen and carbon dioxide, as the process will eliminate methanogenic activity under higher pH (more than 6.5). This pre-treatment could also promote the *Enterobacter* and *Clostridium* family, which can assist in sludge solubilization [30].

Subsequently, when comparing the substrates in the Second Phase, the results show that the source of inoculum and substrate influence the volume of biohydrogen produced. These findings could be due to the different compositions of the substrates [31]. A study

has reported a biohydrogen production rate of 17.05 mL $H_2$/h using heat-treated sewage sludge as inoculum and brewery wastewater as substrate [28]. This value could be influenced by the reactor used (anaerobic baffled reactor) and the retention time of 10.2 h. Meanwhile, Regueira et al. (2020) state that various microorganisms could activate or inactivate hydrogen consumption, resulting in unstable VFA and hydrogen production [32]. This condition is influenced by the different pH used during digestion and inoculum pretreatments conditions. Yossan et al. (2012) reported that *Clostridium* spp. was a dominant species after inoculum heat-shock treatment with POME for biohydrogen production [33] while García-Depraect et al. (2017) found that acetic acid bacteria, such as *Acetobacter* spp., was dominant during the biohydrogen digestion process that converts glucose to acetic acid [34].

As shown in Figure 5, the POME concentration (the organic content) was higher (13 g COD/L) than the Carlsberg (12.5 g COD/L). This result could explain the significant difference in biohydrogen production between the two substrates. POME concentration is slightly higher than that of Carlsberg (12.5 g COD/L) because the former contains cellulose and hemicellulose, which are further degraded by the active microbes in the inoculums to produce biohydrogen, hence generating higher amounts of hydrogen [35,36]. The average biohydrogen production rate is also higher than the First Phase, which uses glucose (6 g/L) as a substrate. Lignocellulosic biomass, a carbohydrate-rich substrate, such as raw POME, could also produce hydrogen by anaerobic bacteria [1,21].

Besides that, Carlsberg sludge used with Carlsberg and F&B effluent yielded a 10% difference in the volume of biohydrogen production rate, despite F&B having approximately six times lower organic content than Carlsberg. Furthermore, using POME sludge with raw POME yielded about nine and eight times more hydrogen than when used with Carlsberg and F&B effluent, despite the COD difference being approximately two and eleven times less in Carlsberg and F&B, respectively. This finding might indicate that brewery wastewater contains inhibitive compounds, limited essential macro- and micronutrients, or a combination of these factors that may affect microbial metabolism [37].

Next, in the Third Phase, the increase of COD concentration from 5 g/L to 10 g/L and 28 g/L COD was accompanied by a 3.4- and 3.8-fold increase in biohydrogen production, respectively. Considering that 5 g/L of glucose was added to amend the COD content to 10 g/L, only a small percentage of the organic content in POME was accessible to HPB. However, 34 mL $H_2$/d was recorded using 28 g/L for raw POME. This finding is not proportional to 5 g/L POME as substrate. Therefore, it is postulated that, when using 28 g/L POME, about 54 mL $H_2$/d could be produced. The reason could be attributed to the substrate types used in this study. More complex substrates, such as raw POME, must be broken down into simple structures for easier access during hydrogen fermentation, allowing the HPB to quickly digest the simple substrates after appropriate pre-treatment [30].

Additionally, raw POME contains 16–100 g/L of total COD, indicating high organic nutrients [38]. The proximate analysis of raw POME from another study also showed that the carbohydrate content in raw POME was significantly high [39]. The high nutrient content in raw POME makes it a potential source to produce biohydrogen using biological treatment. Additionally, the food-to-microbe ratio (F/M) in POME sludge (35 g VSS/L) cultured in raw POME (28 g COD/L) yields an F/M of 0.8. An F/M between 0.5 and 1.0 is reported to be suitable in a batch anaerobic test for food wastes [40], while the 0.5–1.4 range is ideal for high-strength organic wastewater [41]. In addition, this strategy has shown that raw POME from a cooling pond can produce a biohydrogen volume similar to POME (5 g/L) mixed with glucose (5 g/L) from an anaerobic pond.

Based on the obtained results, this study proves that biohydrogen production is feasible using POME sludge with raw POME. The mixture of POME sludge and raw POME would give a pH of approximately 6 that will slightly decrease (around pH 5–5.5) after the reaction due to the production of VFAs. Another approach is maintaining the thermophilic condition during the process [1]. During the acclimated process, adaptation will

occur between the mixed cultures in both inoculum and substrate, thus improving biohydrogen production.

Two types of microorganisms that produce hydrogen are photosynthetic bacteria and anaerobic bacteria. Many studies have reported using mixed cultures of anaerobic bacteria for biohydrogen production [42,43]. Khanal (2004) and Lay (1999) reported that biohydrogen production studies often use heat-treated biological waste as a ready source of hydrogen-producing mixed microflora [44,45]. *Clostridia* are primarily found in mixed cultures enriched from natural environments [45,46]. However, our previous study using mixed cultures from heat-treated POME sludge found that Gram-negative rod bacteria had the highest count ($2.5 \times 10^7$ CFU/mL). The partial 16S rRNA gene sequences confirmed that 95% was similar to *Acetobacter* spp., which predominates in the fermentation process that treats raw POME [16]. Kumar et al. (2018) revealed that, under mesophilic conditions, the most commonly reported HPBs are *Enterobacter* (Gram-negative, non-spore former) and *Clostridium* (Gram-positive, spore former) [47]. These findings proved that both biohydrogen-producing and non-producing microorganisms coexist in a dark fermentative biohydrogen production system. They can be found in various ecosystems as a single strain or communities of different taxa. Nonetheless, the most promising microbes for biohydrogen production via dark fermentation are known to be obligate and facultative anaerobes [47].

## 5. Conclusions

The anaerobic fermentation batch study using anaerobic mixed cultures demonstrated the feasibility of biohydrogen generation utilizing feedstock wastes as substrate. The heat-treatment method used for the selective enrichment of HPB influenced the total biohydrogen volume produced. The heat-treatment method at different temperatures and times positively influenced biohydrogen production. In this study, POME sludge (50 g TSS/L) and raw POME (28 g COD/L) had the highest potential inoculum and substrate concentration for efficient biohydrogen generation (34 mL $H_2$/d). However, using POME as a substrate, Carlsberg sludge could also be a good inoculum source for biohydrogen production. Inoculum heat treatment at 80 °C and 30 min was the best condition for biohydrogen production.

These findings proved that this guideline was successfully implemented to evaluate the performance and potential of the heat-treatment process used for producing biohydrogen from different organic biomass. This study could also be a reference for various communities, such as academia and industrial players, in identifying the most promising inoculums and substrates for biohydrogen production, allowing for future investigation. A substrate pre-treatment could enhance the hydrogen production rate due to the high lignin content. Additional research on the microbial characteristics during inoculum heat treatment and after the incubation period could be performed to determine the most prominent bacteria during the process.

**Author Contributions:** Conceptualization, A.A.Z.; data curation, S.Z.; formal analysis, B.S.Z. and S.Z.; funding acquisition, P.J.K. and H.M.; investigation, S.Z.; methodology, S.Z.; resources, B.S.Z. and S.Z.; supervision, S.I., P.J.K. and H.M.; validation, B.S.Z. and S.Z.; visualization, B.S.Z. and S.Z.; writing—original draft, S.Z.; writing—review & editing, B.S.Z., S.Z., N.S.M., H.M. and P.J.K. All authors have read and agreed to the published version of the manuscript.

**Funding:** This research was funded by the University of Malaya under the Industrial Prototype Grant (grant number RU019D-2014A), and the APC was funded by the UNITEN BOLD grant (J510050002) from Universiti Tenaga Nasional as well as the Highly Cited Research (HCR) Track program, and AAIBE Chair of Renewable Energy grant.

**Institutional Review Board Statement:** Not applicable.

**Informed Consent Statement:** Not applicable.

**Data Availability Statement:** Not applicable.

**Acknowledgments:** We are grateful to Carlsberg (M) and Jugra Palm Oil Mill (now known as Seri Bandar Palm Oil Mill Sdn Bhd) for providing us with samples. We also thank the University of Malaya for allowing us to conduct this research. Finally, all authors would like to express gratitude to Nur Atiqah Aziz and Muhammad Ridzwan Muhammad Ramdan for the knowledge transfer.

**Conflicts of Interest:** The authors declare no conflicts of interest.

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
