# Peer review of "Preliminary Investigation of Different Types of Inoculums and Substrate Preparation for Biohydrogen Production"

_fermentation, doi:10.3390/fermentation9020127_

Round 1

Reviewer 1 Report

Reviewer Comments

I have reviewed the manuscript titled “A study comparing different types of inoculums and substrates toward biohydrogen production” The authors have done a preliminary investigation of different types of inoculums and substrates preparation for biohydrogen production. The research conception and the idea of the study is good. The study has a significant implication for a scale-up of biohydrogen production. The comparative study will also help identify the best conditions for optimum yield of biohydrogen. However, the authors should address the following:

Title of manuscript:

I suggest you re-capture the title of your manuscript as " Preliminary investigation of different types of inoculums and substrates preparation for biohydrogen production”. Consider the rephrase title or react to it in case, you felt it’s not speaking to what you did.

Abstract:

- Re-write the abstract, the word "before” is not a good idea to write an abstract.

- How did you arrive at the best conditions? Did you optimize the study

Introduction:

-Introduction require a re-working

- line 37: recast you cannot write “in interest in”

- line 38-45: Cite the right and updated literature in your introduction

- line 63-66: Discuss briefly the significant critical factors of Mohan et al and the results authors obtained.

- -line 70-71: refer to the pieces of literature by citation?

- The paper objective of line 74-76; should end the introduction

Materials and methods:

-          line 92-93: What did you use to analyse this?

-          line 95-97: Table 1? microbial characterization is missing? especially as regard to inoculum formation or preparation.

-          Section 2.3 headings should read: Preliminarily investigation experiment

-          line 109-111: recast

-          101-111; repetition, you said it earlier

-          line 113-115: How did you reach the optimum?

-          Discuss Figure 2 comprehensively

-          line 120? What’s the basis for the choice of temperature ranges

-          Section 2.3.2: Clearly show the design of the study in table rather than referring to figures

-          line 134-135: Basis for choosing those stated conditions

-          line 145-146: What is the basis and science of those ratios or just trial?

-          Mention this in full at first mention? COD, VSS, TSS

-          Table 2: Since pH is constant, just mention in your explanation. Not necessary in the table

Discussion:

-          The discussion needs to show more of science behind the results and not just observations.

Conclusions:

-          Line 295: start with “the” not “this”

-          The conclusion should reflect the significance of the study to a large community or industrial sector

-          Propose a further or recommendation study as an improvement to further your studies

General comments:

-          Authors should also check the attached pdf of the manuscript for other corrections not captured above.

Author Response

Reviewer #1
I have reviewed the manuscript titled “A study comparing different types of inoculums and substrates toward biohydrogen production” The authors have done a preliminary investigation of different types of inoculums and substrates preparation for biohydrogen production. The research conception and the idea of the study is good. The study has a significant implication for a scale-up of biohydrogen production. The comparative study will also help identify the best conditions for optimum yield of biohydrogen. However, the authors should address the following:
Comment 1: Title of manuscript: I suggest you re-capture the title of your manuscript as " Preliminary investigation of different types of inoculums and substrates preparation for biohydrogen production”. Consider the rephrase title or react to it in case, you felt it’s not speaking to what you did.
Response 1: Dear Reviewer, thank you for the suggestion. We agree on the title suggested.
Author Actions: Title changed to ‘Preliminary investigation of different types of inoculums and substrate preparation for biohydrogen production’.
Comment 2: Abstract: Re-write the abstract, the word "before” is not a good idea to write an abstract.
Response 2: We thank the reviewer for highlighting this concern. The word was removed.
Author Actions: The statement was rephrased to ‘A pre-culture stage is required to obtain robustly dividing cells with high hydrogen (H2) production capabilities.’
Lines 19-20
Comment 3: Abstract: How did you arrive at the best conditions? Did you optimize the study
Response 3: The best conditions were optimized based on one factor at a time method. During the First Phase, Carlsberg sludge was heat treated at different time and temperatures. Next, the best conditions from the First Phase (time, temperature) were selected for the Second Phase. In this phase, different inoculums and substartes were evaluated for biohydrogen production. In the Third Phase, different substrate concentrations were studied using the best heat-treated inoculum (from the Second Phase). Finally, the optimum conditions were based on the best substrate concentration and heat-treated inoculum.
Author Actions: The steps described above was pictured in Figure 2 in the manuscript.
Lines 122-129.
Comment 4: Introduction: Introduction require a re-working
Response 4: We thank the reviewer for the suggestion given. We have made some amendments on the introduction.
Author Action: The introduction has been syntaxed and amended. Lines 50-53, 56-64, 65-70, 71-77, 82-86, 87-94.
Comment 5: Introduction: line 37: recast you cannot write “in interest in”
Response 5: We thank the reviewer for the suggestion given. We have rephrased the statement accordingly.
Author Action: The statement has been rephrased to ‘For the past 15 years, focus on hydrogen production from organic waste has been increasing, fuelling research in the field’.
Lines 36.
Comment 6: Introduction:
line 38-45: Cite the right and updated literature in your introduction
Response 6: We thank the reviewer for the suggestion given. We have replaced and amend several recent literatures as our references.
Author Action: The references were added from [9] – [11].
Comment 7: Introduction:
line 63-66: Discuss briefly the significant critical factors of Mohan et al and the results authors obtained.
Response 7: We thank the reviewer for the suggestion given. We have added some findings from the authors.
Author Action: The significant findings were added between lines 82-86.
Comment 8: Introduction:
line 70-71: refer to the pieces of literature by citation?
Response 8: We thank the reviewer for the question given. It is referring to the citations that we mentioned in the text.
Author Action: We have revised the statement as ‘Therefore, based on the factors described above’ to avoid confusion. Line 87.
Comment 9: Introduction:
The paper objective of line 74-76; should end the introduction
Response 9: We thank the reviewer for the suggestion given. The objectives were restructured accordingly.
Author Action: The content was restructured and syntaxed accordingly. Lines 87-94.
Comment 10: Materials and methods:
line 92-93: What did you use to analyse this?
Response 10: We thank the reviewer for the question given. We clarified the methods used in Section 2.4: Analytical method.
Author Action: The characteristics were analysed as explained in Section 2.4: Analytical method.
Comment 11:
Materials and methods:
line 95-97: Table 1? microbial characterization is missing? especially as regard to inoculum formation or preparation.
Response 11: We thank the reviewer for the question raised. However, we did not measure the microbial characteristics of inoculum in this study. This limitation was explained in Conclusion section as recommendation for future study.
Author Action: Referring the above explanation.
Comment 12: Materials and methods:
Section 2.3 headings should read: Preliminarily investigation experiment
Response 12: We thank the reviewer for the suggestion given. We have amended as suggested.
Author Action: The heading was revised to ‘Section 2.3: Preliminarily investigation experiment’. Lines 121.
Comment 13: Materials and methods: line 109-111: recast
Response 13: We thank the reviewer for the suggestion given. We have rephrased as suggested.
Author Action: We have rephrased the statements for better understanding. ‘As shown in Fig. 2, The best conditions were optimized based on one factor at a time method. During the First Phase, Carlsberg sludge was heat treated at different time and temperatures. Next, the best conditions from the First Phase (time, temperature) were selected for the Second Phase. In this phase, different inoculums and substrates were evaluated for biohydrogen production. In the Third Phase, different substrate concentrations were studied using the best heat-treated inoculum (from the Second Phase). Finally, the optimum conditions were based on the best substrate concentration and heat-treated inoculum. The details of each phase are described below’.
Lines 122-129.
Comment 14: Materials and methods:
101-111; repetition, you said it earlier
Response 14: We thank the reviewer for the information given. We have excluded the repetition statement.
Author Action: Figure 1 had explained the source of raw POME used in this study. Hence, the statement in Lines 101-102 were removed.
Comment 15: Materials and methods: line 113-115: How did you reach the optimum?
Response 15: We thank the reviewer for the question given. The best conditions were selected as explained in Comment 3.
Author Action: The explnataion was added between lines 122-129.
Comment 16: Materials and methods:
Line 120 What is the basis for the choice of temperature ranges?
Response 16: We thank the reviewer for the question given. The choice of the temperature ranges were explained in the text.
Author Action: Temperatures between 80 – 100 ï‚°C between 30 - 120 mins were selected based on studies done by Lin et al. (2011) for 80 Celsius degrees 30 mins [20], Uyub et al. (2017) for 80  Celsius degrees, 120 mins and 100  Celsius degrees, 30 mins [21], and Woo & Song (2010) for 100 ï‚°C, 120 mins [22]. Meanwhile, a heat treatment at 90 ï‚°C for 75 mins was selected as the median between 80 – 100  Celsius degrees and 30 – 120 mins. These statements were added between Lines 135-139.
Comment 17: Materials and methods:
Discuss Figure 2 comprehensively
Response 17: We thank the reviewer for the suggestion given. The details were explained in subsection 2.3.1 until 2.3.3.
Author Action: Detailed explanation were given in subsection 2.3.1 until 2.3.3. Lines 133 – 172.
Comment 18: Materials and methods:
line 120? What’s the basis for the choice of temperature ranges
Response 18: We thank the reviewer for the question given. As explained in Comment 16, temperatures between 80 – 100  Celsius degrees between 30 - 120 mins were selected based on studies done by Lin et al. (2011) for 80  Celsius degrees, 30 mins [15], Uyub et al. (2017) for 80  Celsius degrees, 120 mins and 100 Celsius degrees, 30 mins [16], and Woo & Song (2010) for 100 Celsius degrees, 120 mins [17]. Meanwhile, a heat treatment at 90  Celsius degrees for 75 mins was selected as the median between 80 – 100  Celsius degrees and 30 – 120 mins.
Author Action: We have justified the selection and added between lines 135-139.
Comment 19: Materials and methods:
Section 2.3.2: Clearly show the design of the study in table rather than referring to figures
Response 19: We thank the reviewer for the sugegstion given. However, due to the continuous experiment and using one factor at a time method, we believe that a figure (Figure 2) could give a clearer view of the whole experiment.
Author Action: The design of the study remains a figure (Figure 2).
Comment 20: Materials and methods:
line 134-135: Basis for choosing those stated conditions
Response 20: We thank the reviewer for the question given. Again, as explained in Comment 3, the basis for selecting the best conditions were according to one factor at a time method.
Author Action: We have explained the selection and added between lines 122-129.
Comment 21: Materials and methods:
line 145-146: What is the basis and science of those ratios or just trial?
Response 21: We thank the reviewer for the question given. 5 g/L was chosen as we do not want the bacteria to have a feed shock when we first introduced a different substrate for them. The detail explanation was added in the Discussion part.
Author Action: The discussion was added in Line 310-317.
Comment 22: Materials and methods: Mention this in full at first mention? COD, VSS, TSS
Response 22: We thank the reviewer for the question given. These abbreviations were first mentioned in Table 1.
Author Action: Table 1 defined COD, TSS, and VSS.
Comment 23: Materials and methods: Table 2: Since pH is constant, just mention in your explanation. Not necessary in the table.
Response 23: We thank the reviewer for the suggestion given. As suggested by Reviewer 2 on the same concern, we have removed Table 2 and replaced with Figure 4 for better references for the readers.
Author Action: Table 2 was replaced with Figure 4. pH was removed as suggested.
Comment 24: Discussion: The discussion needs to show more of science behind the results and not just observations.
Response 24: We thank the reviewer for the question given.
Author Action: We have added related discussions on each phases as suggested. The amendments can be found between lines 268-274, 276-280, 304-306, 313-317.
Comment 25: Conclusions: Line 295: start with “the” not “this”
Response 25: We thank the reviewer for the suggestion given. We have revised accordingly.
Author Action: This was changed to ‘The’.
Line 337.
Comment 26: Conclusions: The conclusion should reflect the significance of the study to a large community or industrial sector.
Response 26: We thank the reviewer for the suggestion given. The significance of the study to the community/industries were added between Lines 349-355.
Author Action: The significance of the study was amended in Conlcusion section as suggested.
Comment 26: Conclusions:
Propose a further or recommendation study as an improvement to further your studies
Response 26: We thank the reviewer for the question given. The recommendation was added between lines 349-355.
Author Action: Recommendation was added as suggested in Conlcusion section.
Comment 27: General comments:
Authors should also check the attached pdf of the manuscript for other corrections not captured above.
Response 27: We thank the reviewer for all the suggestions and questions given to improve the manuscript. We have go through the comments as suggested and amend accordingly.
Author Action: Amend accordingly as suggested/commented.

Reviewer 2 Report

1. Introductory section should be expanded. There is no critical analysis in this section. The authors are suggested to evaluate the advantages and disadvantages of various potential assessing methods of hydrogen production by fermentation. In fact, only the last three paragraphs of the introduction are relevant to the article topic. Also, inoculums and substrates for biohydrogen production should be described more broadly. 

2. It would be appropriate to indicate the chemical composition of sludges and substrates in the Materials and Methods section. Also, content of water should be indicated. 

3. Rate of biohydrogen formation over time in a graphical form should be presented in the Results section in order to clear dynamics of biohydrogen production. In addition, it would be better to add more formal (clear) indication of biohydrogen formation dynamics. 

4. It is not clear from the text of the article why there are no consistent patterns correlate temperature and duration of heat treatment to biohydrogen production. 

5. The most potential inoculum and substrate concentration for optimal biohydrogen production and other inoculum and substrate rational parameters for the largest amount of biohydrogen production should be added in the Conclusions section. Also, parameters of thermophilic condition during the process must be specified.

Author Response

Reviewer #2
Comment 1:
Introductory section should be expanded. There is no critical analysis in this section. The
authors are suggested to evaluate the advantages and disadvantages of various potential
assessing methods of hydrogen production by fermentation. In fact, only the last three
paragraphs of the introduction are relevant to the article topic. Also, inoculums and substrates
for biohydrogen production should be described more broadly.
Response 1: Dear Reviewer, thank you for the suggestion. We have added the
information as suggested.
Author Actions: The advantages and disadvantages of various potential assessing
methods of hydrogen production by fermentation were added between lines 50-53 and
56-64. Also, info on substrate selection for dark fermentation was added between lines
56-60, and inoculums for biohydrogen production were added between and 68-70.
Comment 2:
It would be appropriate to indicate the chemical composition of sludges and substrates in the
Materials and Methods section. Also, content of water should be indicated.
Response 2: Dear Reviewer, thank you for the suggestion. The chemical composition
of the sludges and substrates used were already tabulated in Table 1, Materials and
Method section. Since we did not measure the water content of all samples, the value
for POME was cited from our team's publication.
Author Actions: Table 1 displays the compositions of all samples.
Comment 3:
Rate of biohydrogen formation over time in a graphical form should be presented in the Results
section in order to clear the dynamics of biohydrogen production. In addition, it would be better
to add more formal (clear) indication of biohydrogen formation dynamics.
Response 3: Dear Reviewer, thank you for the suggestion. The data in the table were
pictured as a figure for better reference.
Author Actions: Table 2 was replaced with Figure 4 to translate the data better.
Comment 4:
It is not clear from the text of the article why there are no consistent patterns correlate
temperature and duration of heat treatment to biohydrogen production.
Response 4: Dear Reviewer, thank you for the suggestion. Temperatures between 80 –
100 Celsius degrees between 30 - 120 mins were selected based on studies done by Lin et al. (2011)
for 80 Celsius degrees, 30 mins [14], Uyub et al. (2017) for 80 Celsius degrees, 120 mins and 100 Celsius degrees, 30 mins
[15], and Woo & Song (2010) for 100 C, 120 mins [16]. Meanwhile, a heat treatment
at 90 Celsius degrees for 75 mins was selected as the median between 80 – 100 Celsius degrees and 30 – 120
mins. The justifications were added in the text.
Lines 122-126.
Author Actions: We have added the justifications in Subsection 2.3.1: First Phase:
Inoculum Heat Treatments.
Comment 5:
The most potential inoculum and substrate concentration for optimal biohydrogen production
and other inoculum and substrate rational parameters for the largest amount of biohydrogen
production should be added in the Conclusions section. Also, parameters of thermophilic
condition during the process must be specified.
Response 5: Dear Reviewer, thank you for the suggestion. The amendment was made
in the Conclusion section.
Author Actions: Suggestions added at Lines 341-346.

Reviewer 3 Report

The research problem described in the manuscript is current and concerns the production of biohydrogen by anaerobic fermentation using raw palm oil mill effluent (POME) and POME sludge as a substrate and inoculum. The authors suggest that the presented scientific findings are universal and may also be useful in the case of using various organic wastes for the production of biohydrogen.

In general, the technical side of the manuscript does not raise any objections. The introduction is adequate and complete in terms of content selection. Source materials are properly selected and cited extensively. The experimental part was presented specifically and clearly described. The methods used are adequate to solve the presented research problem. The microbiological aspect of the conducted research needs to be supplemented. Unfortunately, the manuscript does not include important information regarding the characteristics of the microorganisms used for fermentation, the number of cells in the fermentation medium, viability, etc. This is a serious shortcoming that needs to be corrected. The authors point out in the discussion that according to Regueira et al. (2020) various microorganisms can activate or deactivate hydrogen consumption, resulting in unstable VFA and hydrogen production. Meanwhile, the microbiological aspects of the conducted research were not presented in the manuscript.

The obtained research results are interesting for cognitive reasons. However, I would expect a specific presentation of the application aspects of the presented research results.

The drawings are clear and of good quality, as are the tables. The discussion of the results is conducted efficiently but does not take into account the microbiological aspects of the conducted research, which must be supplemented.

Author Response

Response 1: Dear Reviewer, thank you for the comments and suggestion. Since the comments are also raised by other reviewers, we have added the information that this study used mixed cultures, hence no specific microbes were selected, and no microbial characteristics were done.
Author Actions: The term was amended in the conclusion section. Line 337.

Round 2

Reviewer 1 Report

Authors have addressed the suggested comments and queries raised

Author Response

Dear Reviewer 1,

We would like to thank you for the comments given.

Reviewer 2 Report

Accept in present form

Author Response

Dear Reviewer 2,

Thank you for the comment given.

Reviewer 3 Report

The authors' response regarding the type of microorganisms used in the fermentation process is insufficient and unacceptable. The description of the conditions of the scientific experiment should enable its reproduction in order to verify the credibility of the scientific findings. The type and physiological state of the microorganisms used for fermentation is a key process parameter having a direct impact on the obtained results. Lack of the above information in the materials and methods section disqualifies the manuscript from a scientific point of view.

Author Response

Response: Dear Reviewer, we apologise for our previous response. We would like to
thank you for the comments and suggestions given. We agree that the scientific
experimental conditions must be addressed/explained, especially on the microbial
analysis/microorganisms involved during dark fermentation process.
Author Actions: We have went through and amended each of Reviewer 3’s comments
and suggestions in Round 1. We have added the bacterial identification and plate count
for untreated POME sludge (inoculum) in Table 1. The
amendment/information/discussion regarding microbial species and count involved in
this study, especially on untreated and heat-treated POME sludge were added in the
Discussion section (Lines 319 – 322 and 325 - 330). The possible microorganism during
dark fermentation process was also mentioned in this study (Lines 447 - 463). Finally,
inoculum heat-shock pre-treatment chosen in this study was justified in Introduction
section (Lines 91 - 92). All the amendments/justifications were highlighted in yellow.